# Triglyceride and Glucose Index as a Screening Tool for Nonalcoholic Liver Disease in Patients with Metabolic Syndrome

**DOI:** 10.3390/jcm11113043

**Published:** 2022-05-28

**Authors:** Anca Maria Amzolini, Mircea-Cătălin Forțofoiu, Anca Barău Alhija, Ionela Mihaela Vladu, Diana Clenciu, Adina Mitrea, Maria Forțofoiu, Daniela Matei, Magdalena Diaconu, Marinela Sinziana Tudor, Elena Simona Micu

**Affiliations:** 1Department of Medical Semiology, Faculty of Medicine, University of Medicine and Pharmacy of Craiova, 200349 Craiova, Romania; amzolinianca@yahoo.com (A.M.A.); anca_barau@yahoo.com (A.B.A.); gvs.2005@yahoo.com (E.S.M.); 2Department of Internal Medicine 2, “Philanthropy” Clinical Municipal Hospital of Craiova, 200143 Craiova, Romania; 3Department of Diabetes, Nutrition and Metabolic Diseases, University of Medicine and Pharmacy of Craiova, 200349 Craiova, Romania; ada_mitrea@yahoo.com; 4Department of Emergency Medicine, University of Medicine and Pharmacy of Craiova, 200349 Craiova, Romania; 5Department of Physiotherapy, University of Medicine and Pharmacy of Craiova, 200349 Craiova, Romania; mateidana30@yahoo.com; 6Department of Pharmacology, University of Medicine and Pharmacy of Craiova, 200349 Craiova, Romania; magdalena.diaconu@umfcv.ro; 7Doctoral School, University of Medicine and Pharmacy of Craiova, 200349 Craiova, Romania; marinelasinziana@yahoo.com

**Keywords:** nonalcoholic liver disease, triglyceride and glucose index, metabolic syndrome, liver biopsy, obesity

## Abstract

Background: Nonalcoholic fatty liver disease (NAFLD) is regarded as a component of metabolic syndrome, which involves insulin resistance (IR) as the primary physiopathological event. The aim of this study was to establish the association between IR, assessed using the triglyceride and glucose index (TyG), and histopathological features of NAFLD lesions. Methods: The study included 113 patients with metabolic syndrome. Fasting plasma glucose (FPG), fasting lipid profiles and liver enzymes were measured. IR was assessed by the TyG index. Liver biopsy was performed for assessment steatosis and fibrosis. Results: the TyG index had a mean value of 8.93 ± 1.45, with a higher value in the patients with overweight (*p* = 0.002) and obesity (*p* = 0.004) characteristics than in the patients with normal weight. The TyG index mean value was 8.78 ± 0.65 in subjects without NASH, 8.91 ± 0.57 in patients with borderline NASH and 9.13 ± 0.55 in patients with definite NASH. A significant difference was found between subjects without NASH and the ones with definite NASH (*p* = 0.004), as well as in patients with early fibrosis vs. those with significant fibrosis. The analysis of the area under the ROC curve proved that the TyG index is a predictor of NASH (*p* = 0.043). Conclusion: the TyG index is a facile tool that can be used to identify individuals at risk for NAFLD.

## 1. Introduction

Nonalcoholic fatty liver disease (NAFLD) covers a spectrum of liver lesions (steatosis, non-alcoholic hepatosteatosis—NASH, fibrosis, cirrhosis) that appear in the absence of alcohol consumption or minimal alcohol consumption (under 20 g of pure alcohol for females and under 30 g of pure alcohol for males) in the absence of other causes of liver disease [1].

Many studies that prove the association of NAFLD with insulin resistance (IR), obesity, high blood pressure and dyslipidemia regard NAFLD as a component of metabolic syndrome [2].

IR is considered the primary event in the physiopathology of NAFLD/NASH in many studies discussing theories related to NAFLD pathogenesis. There is a bidirectional relationship between the liver and IR: IR and secondary hyperinsulinism lead to liver disease through lipid accumulation, and the liver disease exacerbates IR, transforming impaired glucose tolerance into overt diabetes mellitus through insulin metabolism impairment [3].

Liver biopsy is considered the “gold standard” for the diagnosis of liver lesions in NAFLD. Although the utility of liver biopsy, which establishes the disease subtypes, appreciates the severity and the progression of the disease, as well as certifies the liver fibrosis, it is universally recognized that this method is associated with a series of disadvantages and problems, which have led to the replacement of liver biopsy with other non-invasive procedures [4].

Even though there is an increasing interest in the study of non-invasive diagnostic methods in order to identify imaging techniques and serological markers that correlate with steatohepatitis and liver fibrosis, there is no specific biomarker that can be used for the diagnosis of NAFLD. Triglyceride accumulation and IR are the hallmarks of NAFLD; therefore, the triglyceride and glucose index (TyG) may predict the subsequent occurrence of NAFLD in later life. Studies showed that there is a strong and positive association between TyG and risk of NAFLD in different populations [5,6].

Other studies showed that the TyG index predicted NAFLD better than the homeostasis model assessment of insulin resistance (HOMA-IR), the most frequently used formula that assesses IR in clinical practice [7]. The TyG index was also positively related to the severity of liver steatosis and the presence of liver fibrosis evaluated with transient elastography in patients with NAFLD [8,9].

Taking into account these data, the aim of this study was to establish the association between IR, assessed using the TyG index, and characteristic NAFLD lesions evaluated by histopathological examination.

## 2. Materials and Methods

### 2.1. Participants

We conducted an epidemiological, cross-sectional, non-interventional study over a period of 3 years (2017–2019) that included a sample of 113 subjects out of 345 patients aged over 18 years old, diagnosed with metabolic syndrome, in whom liver biopsy was performed. Based on body mass index (BMI), the subjects were divided into 3 groups: Group 1—14 patients with normal weight; Group 2—49 patients with overweight; Group 3—50 patients with obesity.

The diagnosis of metabolic syndrome was established using the harmonized metabolic syndrome criteria [10], if the patients met 3 of the following 5 criteria: (1) waist circumference above 94 cm in males and 80 cm in females; (2) elevated triglycerides (≥150 mg/dL) or drug treatment for elevated triglycerides; (3) reduced HDL-cholesterol (<40 mg/dL in males and <50 mg/dL in females) or drug treatment for reduced HDL-cholesterol; (4) elevated blood pressure (systolic ≥ 130 and/or diastolic ≥ 85 mm Hg) or drug treatment for hypertension and (5) elevated fasting plasma glucose (≥100 mg/dL) or personal history of prediabetes or diabetes.

The following exclusion criteria were used: pregnant or breastfeeding females, significant alcohol consumption (more than 30 g/day in males and more than 20 g/day in females), patients diagnosed with hepatitis B, C and D virus infections, patients with autoimmune liver disease, other chronic liver diseases (i.e., primary biliary cirrhosis, Wilson’s disease, hereditary hemochromatosis), patients treated with hepatotoxic drugs (methotrexate, amiodarone, corticotherapy, chemotherapy, hormone therapy) and patients that refused the liver biopsy. Informed consent was signed, in full knowledge of the facts, by each participant in the study, after all the aspects necessary to make a decision for or against enrolling in the study were communicated.

The study was conducted in accordance with the ethical principles stipulated in the Helsinki Declaration, in accordance with good clinical practice, respecting the right to integrity, confidentiality and the option of the subject to withdraw from the study at any time.

### 2.2. Data Collection

Data collected from the patients enrolled in the study included demographic characteristics, medical history, family history, smoking and drinking status, clinical findings and laboratory reports which were recorded accordingly in a data collection sheet. The patients’ relevant physical examinations were performed and anthropometry data (weight and height) were collected. BMI was calculated as weight (in kilograms)/height squared (in meters). BMI was classified according to the World Health Organization (WHO) for white individuals [11] into normal weight (BMI < 25 kg/m^2^), overweight (BMI ≥ 25 kg/m^2^ and < 30 kg/m^2^) and obese (BMI ≥ 30 kg/m^2^). In all the patients, we evaluated waist circumference (WC) and blood pressure (BP), which was measured 3 times using an automated sphygmomanometer after the participants were relaxed and seated for more than 10 min; the mean of the 3 determinations was recorded in the data collection sheet.

Fasting plasma glucose (FPG), fasting lipid profiles and liver enzymes were the laboratory tests that were performed in this study. After 8 to 12 h overnight fasting, 4.0 mL venous blood samples were collected in a plain tube from each patient following standard procedure. The tubes were labeled with the identification numbers of the subjects. All the blood samples were kept in vertical position for 30 min at room temperature (22–24 °C). Centrifugation (around 5000 rpm) for 5–10 min at room temperature was used to separate serum which was afterwards preserved at −20 °C until further analysis. The level of fasting serum triglyceride was determined by the enzymatic colorimetric method, and the FPG was measured by the glucose oxidase method [12]. We assessed insulin resistance (IR) using the triglyceride and glucose index (TyG), which was calculated by the formula: TyG = ln [fasting triglycerides (mg/dL) × FPG (mg/dL)/2] [11].

Liver biopsy, using the Menghini technique, was performed in order to evaluate the presence of liver steatosis, NASH and fibrosis. Liver tissue fragments were fixed in 10% formalin, then embedded in paraffin and sectioned at the microtome. The tissue slides were stained with hematoxylin-eosin, van Gieson, Gomori and Masson trichrome. Histologic grading and staging were performed according to Kleiner’s and Brunts’s classification (2005) [13]. According to this classification, depending on the severity of the lesions, the following anatomopathological findings are quantified: steatosis lesions are scored 0–3 (0: <5%; 1: 5–33%; 2: >33–66%; 3: >66%); lobular inflammation scoring ranges 0 to 3 (0: absent; 1: 2 foci/200 × field; 2: 2–4 foci/200 × field; 3: >4foci/200 × field); hepatocyte ballooning ranges 0 to 2 (0: absent; 1: a few balloon cells; 2: many balloon cells/important ballooning), while fibrosis score range 0 to 4 (0: absent; 1: perisinusoidal or periportal fibrosis, 1A: mild, zone 3, perisinusoidal fibrosis; 1B: moderate, zone 3, perisinusoidal fibrosis; 1C: portal/periportal fibrosis; 2: perisinusoidal and portal/periportal fibrosis; 3: bridging fibrosis; 4: cirrhosis) [13].

NAFLD Activity Score (NAS) ranges from 0 to 8 and represents the sum of scores for steatosis, hepatocyte ballooning and lobular inflammation. NAS scores of 0–2 are not considered diagnostic of NASH, scores of 3–4 are considered “borderline NASH”, while scores ≥5 are considered “definite NASH” [14].

### 2.3. Statistical Analysis

Demographic and clinical data were expressed as means with standard deviations and counts with percentages for continuous and discrete variables, respectively. Descriptive statistics was presented, and differences between the groups were compared, using one-way ANOVA for continuous data and the chi-square test for categorical data. The correlations were assessed using Pearson’s analysis. Multiple logistic regression analysis was used to adjust for covariates. All the statistical tests were regarded as statistically significant, with a *p*-value less than 0.05 (two-sided). Data were analyzed using the Statistical Package for the Social Sciences software, version 26.0 (SPSS Inc., Chicago, IL, USA). Continuous variables were tested for normal distribution using the Shapiro–Wilk test. The differences between the 3 BMI categories were analyzed using the ANOVA test for variables with normal distribution and using Kruskal–Wallis test for variables with abnormal distribution. Continuous variables with normal distribution were presented as mean ± standard deviation (SD), while variables with abnormal distribution were presented as median (interquartile range (IQR)).

## 3. Results

The patients enrolled in the study (31 males and 72 females) had a median age of 53 [20] years old and a median BMI of 29 [6] kg/m^2^. A total of 43 of the studied patients (12 males and 31 females) had the diagnosis of type 2 diabetes. FPG had a median value of 101 [41] mg/dL and the median value of the triglycerides was 156 [117] mg/dL, while the ALT median value registered in the study was 35 [31] U/L. The characteristics of the studied patients are summarized in Table 1.

In our study, the TyG index had a mean value of 8.93 ± 1.45. The comparative analysis of TyG between the three study groups showed a higher value in the patients with overweight and obesity than in the patients with normal weight, the difference being statistically significant both when we compared subjects with normal weight and overweight (*p* = 0.002) as well as between subjects with normal weight and obesity (*p* = 0.004), but there was no statistical significance regarding the TyG value between subjects with overweight and obesity (*p* = 0.549), as is presented in Table 2.

The degree of the liver disease (steatosis, inflammation, ballooning and fibrosis) within the three study groups was analyzed, and the NAS was calculated accordingly.

Liver steatosis was identified in 71.43% of the patients with normal weight and in 91.84% of the patients with overweight, while all the patients with obesity presented liver steatosis (Figure 1a). The statistical analysis showed a significant difference regarding the presence of liver steatosis between the patients with normal weight and overweight (*p* < 0.005), as well as between the patients with normal weight and obesity (*p* < 0.005). Furthermore, we also observed a statistically significant difference between the patients with overweight and obesity (*p* = 0.037).

Liver inflammation was identified in 35.71% of the subjects with normal weight, in 57.14% of the subjects with overweight and in 60% of the patients with obesity (Figure 1b), the difference reaching statistical significance between the subjects with normal weight and overweight (*p* < 0.025) and subjects with overweight and obesity (*p* < 0.01). There was no statistically significant difference regarding inflammation between patients with overweigh and obesity (*p* = 0.773).

Ballooning was observed in 28.57% of the individuals with normal weight, in 48.98% of the patients with overweight and in 56% of the patients with obesity (Figure 1c), with a statistically significant difference between the normal weight and the overweight groups (*p* < 0.029) and between the normal weight and obesity groups (*p* = 0.003). There was no statistically significant difference between overweight versus obesity groups (*p* = 0.483).

Fibrosis was present in 35.71% of the subjects with normal weight, in 51.02% of the patients with overweight and in 60% of the patients with obesity (Figure 1d), reaching a statistically significant difference when we compared patients with normal weight and obesity (*p* = 0.01) but with no statistically significant difference between the normal weight versus overweight groups (*p* = 0.111) as well as in overweight versus obesity groups (*p* = 0.367).

NASH was calculated within the three study groups dividing the patients into three categories: not diagnostic of NASH, borderline NASH and definite NASH. Definite NASH was reported in 14.29% of the patients with normal weight, 32.65% of the patients with overweight and in 30% of the patients with obesity. Borderline NASH was observed in 35.71% of the patients with normal weight, 32.65% in the subjects with overweight and 32% in the subjects with obesity (Figure 2). The statistical analysis showed significant differences when we compared the subjects with normal weight with both subjects with overweight (*p* < 0.05) and obesity (*p* < 0.05). There was no statistically significant difference regarding NASH categories between subjects with overweight and obesity (*p* = 0.717).

The TyG index was analyzed in its relationship with the presence of liver steatosis, showing mean values of 8.85 ± 0.75 in the subjects without liver steatosis and 8.93 ± 0.6 in the subjects with liver steatosis but not reaching statistical significance (*p* = 0.708), as is shown in Figure 3a.

When we studied the relationship between TyG index and liver inflammation, we identified mean values of 8.87 ± 0.68 in subjects without liver inflammation and 8.97 ± 0.55 in patients with liver inflammation, not reaching statistical significance (*p* = 0.407), as presented in Figure 3b.

Regarding the association between TyG index and hepatocyte ballooning, we found a mean value of 8.81 ± 0.61 in the subjects that did not present hepatocyte ballooning, while the subjects with hepatocyte ballooning had a mean TyG index of 9.04 ± 0.58, the difference being statistically significant (*p* = 0.038), as can be observed in Figure 3c.

TyG had a mean value of 9.01 ± 0.58 in the patients presenting liver fibrosis, while the subjects without liver fibrosis had a mean TyG value of 8.82 ± 0.62, the difference not reaching statistical significance (*p* = 0.092). When we compared the TyG index in patients with significant liver fibrosis (F2, F3, F4) with those with early fibrosis (F0, F1), we observed a statistically significant difference of 9.33 ± 0.85 vs. 8.82 ± 0.61 (*p* = 0.001), as is presented in Figure 3d.

The associations between TyG index and NASH categories were also studied. We found a mean TyG index value of 8.78 ± 0.65 in subjects without NASH, 8.91 ± 0.57 in patients with borderline NASH and 9.13 ± 0.55 in patients with definite NASH, but the only statistically significant difference was found between subjects without NASH and the ones with definite NASH (*p* = 0.004), as presented in Figure 4.

The analysis of the area under the ROC curve showed a value of 0.635 (Figure 5), proving that the TyG index is an independent predictor for NASH (*p* = 0.043); however, this value is not high enough to be used in order to determine a cut-off value of the TyG index for the presence of NASH. Furthermore, we observed that TyG is an independent predictor for significant fibrosis (F2, F3, F4) when we analyzed the TyG index ROC curve (*p* = 0.001), with an area under the curve of 0.721. However, we could not establish a cut-off value of the TyG index that could predict the presence of fibrosis F2, F3 and F4.

## 4. Discussion

NAFLD is considered a major public health problem worldwide. Obesity, type 2 diabetes mellitus and dyslipidemia, in the absence of alcohol consumption, are the most important risk factors for NAFLD, with IR as the main pathogenic mechanism.

In the medical literature, the prevalence of liver steatosis in the subjects with obesity is estimated at 73–92%, reaching 100% in subjects with diabetes and morbid obesity [15]. In our study, steatosis had a frequency of 92.03%, being identified in 71.43% of the study subjects with normal weight, in 91.84% of the subjects with overweight and in all the subjects with obesity (100%). Our results, which show a high percentage of liver steatosis in the patients with normal weight, can be explained by the inclusion criteria used in this study, which enrolled only subjects diagnosed with metabolic syndrome. All the subjects that presented normal weight had an abdominal circumference above the normal value [16], suggestive of visceral obesity, which implies triglycerides accumulation in the liver [10]. Furthermore, abdominal obesity is considered a risk factor for NAFLD, even in subjects with BMI within the normal range [17].

Regarding NASH, the prevalence of this disorder is estimated at 2–7% of the general population and 34–40% in patients with abnormal liver enzymes, but in the absence of viral hepatitis markers [10], reaching 37% in patients with morbid obesity [18]. According to data published up until the present, in 12–40% of the subjects, liver steatosis evolves into steatohepatitis and incipient fibrosis within 8 to 13 years [19]. In our study, NASH had a frequency of 32.69% in the subjects with liver steatosis, while liver fibrosis had a frequency of 58.65%.

The TyG index, which uses regular blood tests (FPG, serum triglycerides) is a non-invasive marker of IR as well as a good predictor for NAFLD [20,21]. Studies showed an association of liver steatosis with TyG values above 8.5–8.85 [5,6,22,23]. The mean TyG value reported in our study is 8.93 ± 1.45, with no statistically significant difference between subjects that presented liver steatosis and the ones that did not present this type of liver disorder. Although we found a small percentage of subjects that did not present liver steatosis, the mean value of TyG in these patients was 8.85 ± 0.75, higher than the cut-off presented in previous studies. However, when interpreting the results of our study, it must be taken into consideration that all the patients enrolled in the study were diagnosed with metabolic syndrome, a disorder that is associated with IR and an increased TyG index. Furthermore, the lack of correlation between our results and the data reported in the medical literature, associating TyG with both the presence and the severity of liver steatosis [22], can be explained by different diagnostic methods, as most of the published studies assessed liver steatosis using non-invasive methods, while the diagnosis of liver steatosis in our study was established after liver biopsy and histopathological examination.

Regarding the TyG index as a marker of NASH, in this study we found statistically significant differences (*p* = 0.004) between TyG index values in patients without NASH and definite NASH according to NAS, calculated using histopathological criteria. The analysis of the area under the ROC curve showed a value of 0.635 (*p* = 0.043); however, this value was not high enough to establish a cut-off value that can be used a marker of NASH. Further studies including a higher number of patients are needed in order to evidence a cut-off value that could be used in clinical practice.

In our study, liver fibrosis was met in 53.09% of the studied patients. The frequency of liver fibrosis in patients with normal weight but with abdominal obesity according to waist circumference was 37%; however, the degree of fibrosis in these patients was not higher than F1. The TyG index mean values varied between 8.82 ± 0.62 in patients without liver fibrosis and 9.01 ± 0.58 in patients with liver fibrosis, regardless of the degree of fibrosis, the difference not reaching statistical significance. The studies reporting up until the present a correlation between TyG and liver fibrosis have assessed this parameter through non-invasive methods [23,24,25], such as scoring systems or transient elastography—fibroscan.

Furthermore, Masarone et al. [26] observed that steatohepatitis, diagnosed using liver biopsy, represents the sole feature of liver damage in type 2 diabetes. This observation confirms the hypothesis that IR status and type 2 diabetes increase the risk of advanced fibrosis with consequent worsening of hepatic outcomes; the authors of the study recommended routine follow-up ultrasound examination and liver biopsy in patients with type 2 diabetes at diagnosis as a measure to predict overall survival and to evaluate the risk of liver cirrhosis, hepatocellular carcinoma and cardiovascular disease in these patients [26]. Moreover, in patients with type 2 diabetes, it was shown that the reduction of IR through the pharmacological eradication of hepatitis C virus infection by direct-acting antivirals leads to both a reduction in the onset of type 2 diabetes [27] as well as a decrease in clinical expressions of atherosclerosis [28,29].

The main limitation of our study is its transversal design that does not allow the definition of a cause–effect relationship between the TyG index and the histological changes observed on the liver biopsy. Another limitation is the relatively small number of patients studied. On the other hand, the evaluation of NAFLD lesions using liver biopsy (the gold standard for defining the precise histological alterations of NAFLD) represents the strength of this research and balances the relatively small number of the patients included in the study.

## 5. Conclusions

Our study showed a correlation between IR assessed using the TyG index and histopathological components of NAFLD, especially NASH, and significant fibrosis (F2, F3, F4); however, it was not able to identify a TyG cut-off value suggestive of steatohepatitis or fibrosis. The TyG index can be considered a facile tool used to identify individuals at risk for NAFLD. However, prospective studies are needed in order to evaluate the utility of the TyG index for the progression of liver lesions, facilitating the selection of patients in whom early prevention measures could prevent the evolution of liver steatosis.

## Figures and Tables

**Figure 1 jcm-11-03043-f001:**
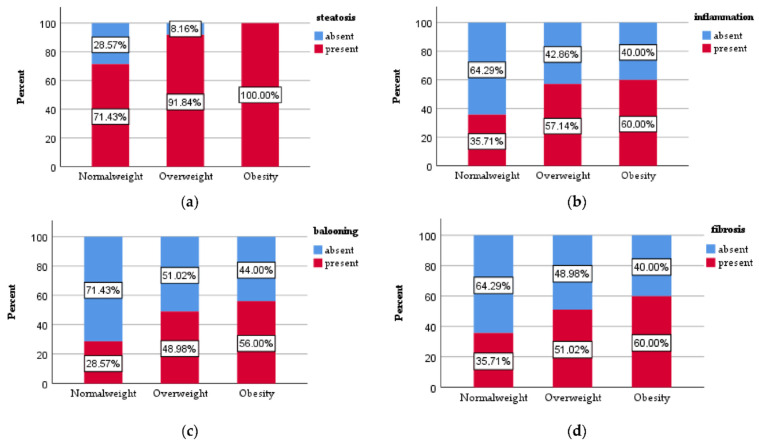
The distribution of (**a**) liver steatosis, (**b**) liver inflammation, (**c**) hepatocyte ballooning and (**d**) fibrosis in the study group.

**Figure 2 jcm-11-03043-f002:**
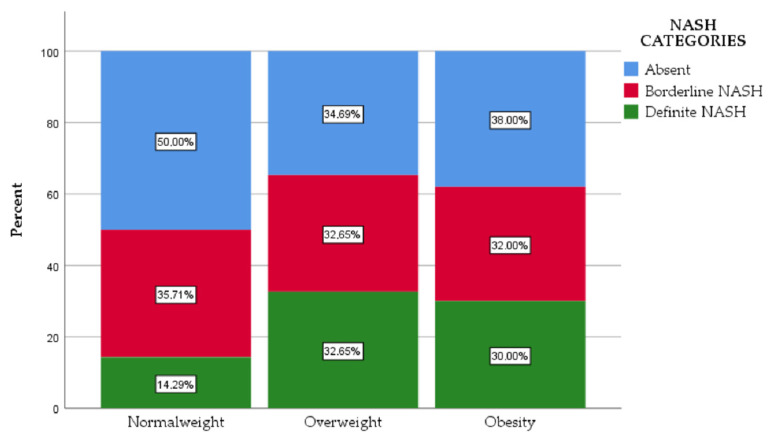
The distribution of the study groups according to NASH category.

**Figure 3 jcm-11-03043-f003:**
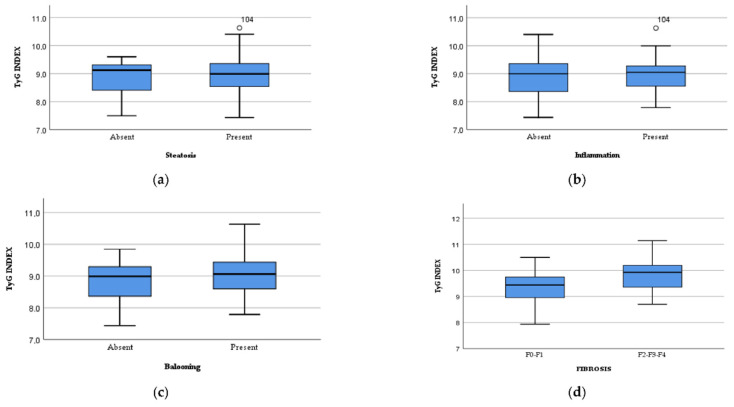
The study of the TyG index in relationship to (**a**) the presence of liver steatosis, (**b**) the presence of liver inflammation, (**c**) the presence of hepatocyte ballooning and (**d**) the degree of liver fibrosis.

**Figure 4 jcm-11-03043-f004:**
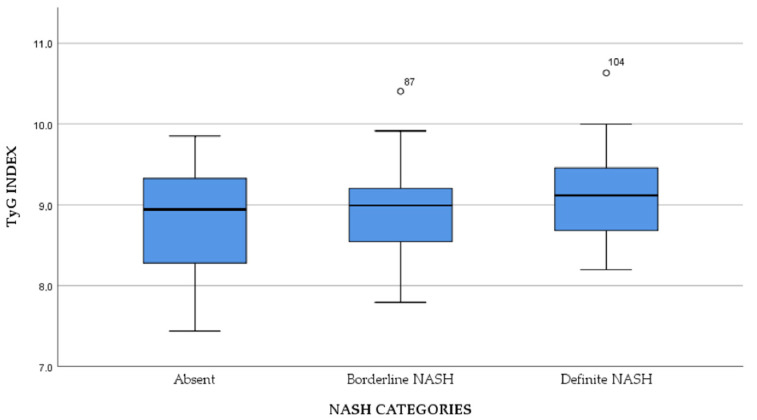
The study of TyG index in relationship with NASH categories.

**Figure 5 jcm-11-03043-f005:**
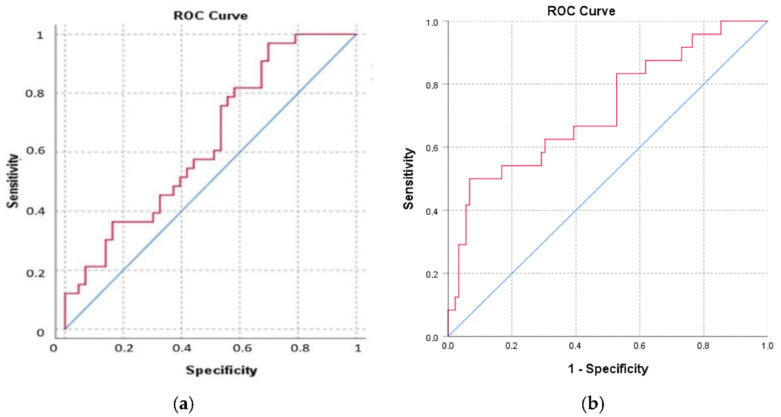
The area under the ROC curve for the TyG index as (**a**) a predictor of NASH; (**b**) a predictor of significant fibrosis (F2, F3, F4).

**Table 1 jcm-11-03043-t001:** Characteristics of the study patients.

Characteristics	Normal Weight	Overweight	Obesity	*p*
Gender (%)				
Male (*n* = 41)	14.6	53.7	31.7	-
Female (*n* = 72)	11.1	37.5	51.4	
Age (years) *	55 [20]	51 [20]	56 [21]	0.016
Waist circumference (cm) †				
Male	101 ± 7.34	102.23 ± 6.1	115.85 ± 6.95	0.001
Female	82.38 ± 1.68	97 ± 9.84	110.57 ± 9.61	
FPG (mg/dL) *	91 [20]	97 [43]	108 [44]	0.032
Triglycerides (mg/dL) *	111 [98]	171 [109]	163 [123]	0.102
Hypertension (%)	20	42.9	54.8	0.317
HDL (mg/dL) *	47 [20]	47 [18]	45 [16]	0.759
ALT (U/L) *	30 [29]	29 [24]	42 [28]	0.024

The data are presented as percentage, median [IQR] or mean ± SD. * Continuous variables with abnormal distribution (presented as mean ± SD). † Continuous variables with normal distribution (presented as median [IQR]).

**Table 2 jcm-11-03043-t002:** The average value of the TyG index in the 3 groups.

GROUP	TyG Index
Normal weight	8.46 ± 0.61
Overweight	8.93 ± 0.55
Obesity	9.05 ± 0.60

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
