# Peer review of "Triglyceride and Glucose Index as a Screening Tool for Nonalcoholic Liver Disease in Patients with Metabolic Syndrome"

_jcm, 2022, doi:10.3390/jcm11113043_

Round 1

Reviewer 1 Report

In this interesting paper the authors observed a correlation between IR assessed using TyG index and histopathological components of NAFLD, especially NASH. The manuscript is quite original. However, this reviewer raises some issues.

1- A paragraph on the limitations of the study is missing at the end of the discussion. The main limitation of the study is its transversal design which does not allow to define a cause-effect relationship between the histological changes observed on the liver biopsy and the TyG index. Another limitation is the relatively small number of patients studied. On the other hand, the biopsy study, which is the gold standard for defining the precise histological alterations of NAFLD, represents the strength of the study and balances the relatively small number of the sample studied.

2- In conclusion, the authors state that the TyG index is as easy as it is useful for identifying individuals at risk for NAFLD, but not for the progression of liver damage. How can a cross-sectional study evaluate the progression of liver damage? This statement must necessarily be correct.

3- The number of patients recruited into the study is missing both in the abstract and in Table 2.

4- Neither in the text nor in Table 2 is the number of diabetic patients recruited in the study indicated. This information must be added.

5- Notably, it was recently observed by liver biopsy that steatohepatitis represents the sole feature of liver damage in type 2 diabetes (PLoS One. 2017 Jun 1;12(6):e0178473. doi: 10.1371/journal.pone.0178473.). This observation confirms the hypothesis that T2D and IR status increase the risk of advanced fibrosis, with the consequent worsening of hepatic outcomes. This important issue should be commented on in the discussion, and the above reference should be added.

6- As stated by authors, IR is the strongest pathophysiological link between NAFLD and Metabolic Syndrome. Recent studies have shown that the reduction of IR through the pharmacological eradication of HCV by direct-acting antivirals leads to both a reduction in the onset of type 2 diabetes (Diabetes, Obesity and Metabolism, 2020, 22(12):2408–2416. doi: 10.1111/dom.14168) and clinical expressions of atherosclerosis (Atherosclerosis, 2020, 296:40–47. doi: 10.1016/j.atherosclerosis.2020.01.010 - Nutrition, Metabolism & Cardiovascular Diseases, 2021, 31, 2345e2353. doi: 10.1016/j.numecd.2021.04.016). These interesting issues as well as the above references deserve to be commented in discussion by the authors.

7- The legend of figures 1-5 should define how the 3 BMI groups are divided. Furthermore, in figures 1-4, steatosis must be indicated as present or absent, and not as 0 or 1. In figure 5, NASH mist be divided in not diagnostic, borderline and definite NASH, and not as 0,1 and 2.

8- Moreover, in figures 6-9, the biopsy parameters must be indicated as absent or present and not with the numbers 0 and 1. In figure 10, the three NASH categories must be well defined and not simply indicated with the numbers 0, 1 and 2.

9- A linguistic revision by a native English speaker is required.

Author Response

Point 1: A paragraph on the limitations of the study is missing at the end of the discussion. The main limitation of the study is its transversal design which does not allow to define a cause-effect relationship between the histological changes observed on the liver biopsy and the TyG index. Another limitation is the relatively small number of patients studied. On the other hand, the biopsy study, which is the gold standard for defining the precise histological alterations of NAFLD, represents the strength of the study and balances the relatively small number of the sample studied.

Response 1: Thank you for pointing this out. We have added the study strenghts and limitations in the Discussion section of the manuscript.

Point 2: In conclusion, the authors state that the TyG index is as easy as it is useful for identifying individuals at risk for NAFLD, but not for the progression of liver damage. How can a cross-sectional study evaluate the progression of liver damage? This statement must necessarily be correct.

Response 2: We thank the reviewer for the comment. We have repharsed this statement.

Point 3: The number of patients recruited into the study is missing both in the abstract and in Table 2.

Response 3: Thank you for the observation. We have added the missing data.

Point 4: Neither in the text nor in Table 2 is the number of diabetic patients recruited in the study indicated. This information must be added.

Response 4: Thank you for the comment. We have added the missing data.

Point 5: Notably, it was recently observed by liver biopsy that steatohepatitis represents the sole feature of liver damage in type 2 diabetes (PLoS One. 2017 Jun 1;12(6):e0178473. doi: 10.1371/journal.pone.0178473.). This observation confirms the hypothesis that T2D and IR status increase the risk of advanced fibrosis, with the consequent worsening of hepatic outcomes. This important issue should be commented on in the discussion, and the above reference should be added.

Response 5: Thank you for the suggestion. We have added the required comments in the Discussion section of the manuscript and we have included the articol you mentiond in the reference list.

Point 6: As stated by authors, IR is the strongest pathophysiological link between NAFLD and Metabolic Syndrome. Recent studies have shown that the reduction of IR through the pharmacological eradication of HCV by direct-acting antivirals leads to both a reduction in the onset of type 2 diabetes (Diabetes, Obesity and Metabolism, 2020, 22(12):2408–2416. doi: 10.1111/dom.14168) and clinical expressions of atherosclerosis (Atherosclerosis, 2020, 296:40–47. doi: 10.1016/j.atherosclerosis.2020.01.010 - Nutrition, Metabolism & Cardiovascular Diseases, 2021, 31, 2345e2353. doi: 10.1016/j.numecd.2021.04.016). These interesting issues as well as the above references deserve to be commented in discussion by the authors.

Response 6: Thank you for the suggestion. We have added the required comments in the Discussion section of the manuscript and we have included the articol you mentiond in the reference list.

Point 7: The legend of figures 1-5 should define how the 3 BMI groups are divided. Furthermore, in figures 1-4, steatosis must be indicated as present or absent, and not as 0 or 1. In figure 5, NASH mist be divided in not diagnostic, borderline and definite NASH, and not as 0,1 and 2

Response 7: Thank you for the comment, we have made the suggested changes.

Point 8: Moreover, in figures 6-9, the biopsy parameters must be indicated as absent or present and not with the numbers 0 and 1. In figure 10, the three NASH categories must be well defined and not simply indicated with the numbers 0, 1 and 2

Response 8: Thank you for pointing this out. We made the necessary changes.

Point 8: A linguistic revision by a native English speaker is required.

Response 9: Thank you for pointing this out. We have revised the manuscript.

Reviewer 2 Report

In the present manuscript, the authors investigated the association between TyG index and the histological features of NAFLD/NASH in a well characterized cohort of patients with metabolic syndrome. 

The availability of liver biopsy is a strenght of the study. Results, despite quite novel, are interesing and worthy for consideration by Journal of Clinical Medicine. However, some aspects need to be clarified and improved.

Below the specific comments:

1) Patients included in the study are those with metabolic syndrome. I suggest to provide a clear definition (and relevant reference) in the methods section indicating all the  medical conditions assessed for the definition of metabolic syndrome. Furthemore, provide HDL data in table 1.

2) Statistical analysis. Continuous variables were reported as mean +/- SD, and data were analyzed using parametric tests (i.e. one-way ANOVA, Pearson correlation). Rarely, the variables considered are normally distributed (for instance see: ALT; mean 44.41 +- 39.03 U/L). I recommend to provide continuous variables as median and interquartile range (IQR), and use non-parametric statistical tests, such as Kruskal-Wallis, Mann-Whitney and Spearman correlation. Furthemore, provide age, FPG, Triglycerides, and ALT values without decimals, and BMI with 1 decimal value.

3) Authors analyzed BMI groups and TyG values variation according to 3 different NASH categories: no NASH, borderline NASH, and definite NASH. Considering that NASH is histological defined as the joint presence of inflammation, ballooning and steatosis, i recommend to define the population in NASH and non-NASH and reanalyze data accordingly. Furthemore, Figures 1-5 could be merged in a single panel as well as figures 6-10.

4) Figure 4 and figure 9. It is unclear to me if the authors considered in the fibrosis group all stages of liver fibrosis (from F1 to F4). I may suggest to compare F01 vs F234 (significant fibrosis) and/or F012 vs F34 (severe fibrosis).

5) Diagnostic accuracy of TyG. Given the availability of liver fibrosis staging, I suggest the authors to investigate the diagnostic accuracy of TyG index for the detection of significant fibrosis (F01 vs F234) and severe fibrosis (F012 - F34).

6) Please provide footnotes for tables and figures, indicating the statistical test used, abbreviations, and all relevant information useful to readers.

7) I suggest to provide subheadings in the Materials and Methods section. 

Author Response

Point 1: Patients included in the study are those with metabolic syndrome. I suggest to provide a clear definition (and relevant reference) in the methods section indicating all the  medical conditions assessed for the definition of metabolic syndrome. Furthemore, provide HDL data in table 1.

Response 1: We absolutely agree, therefore we have added the definition of metabolic syndrome used in our study and we have added HDL levels in table 1.

Point 2: Statistical analysis. Continuous variables were reported as mean +/- SD, and data were analyzed using parametric tests (i.e. one-way ANOVA, Pearson correlation). Rarely, the variables considered are normally distributed (for instance see: ALT; mean 44.41 +- 39.03 U/L). I recommend to provide continuous variables as median and interquartile range (IQR), and use non-parametric statistical tests, such as Kruskal-Wallis, Mann-Whitney and Spearman correlation. Furthemore, provide age, FPG, Triglycerides, and ALT values without decimals, and BMI with 1 decimal value.

Response 2: We thank the reviwer for the comments. We have revised the ststistical analysis acoording to the reviwer’s suggestions.

Point 3: Authors analyzed BMI groups and TyG values variation according to 3 different NASH categories: no NASH, borderline NASH, and definite NASH. Considering that NASH is histological defined as the joint presence of inflammation, ballooning and steatosis, i recommend to define the population in NASH and non-NASH and reanalyze data accordingly. Furthemore, Figures 1-5 could be merged in a single panel as well as figures 6-10.

Response 3: We thank the reviwer for the comments. As we have ststed in the Material and methods, we have used the Brunt criteria for defining NASH, therfore we have to analyze the data according to the 3 chategories included in this classification. However, we agree that the figures 1-4 and 6-9 should be merged, therefore, we have made this modifications.

Point 4:  Figure 4 and figure 9. It is unclear to me if the authors considered in the fibrosis group all stages of liver fibrosis (from F1 to F4). I may suggest to compare F01 vs F234 (significant fibrosis) and/or F012 vs F34 (severe fibrosis).

Response 4: Thank you for the comment, we have analyzed the data according to your suggestions, observing statistically significant differences between early fibrosis stages (F0, F1) and significant fibrosis (F2, F3, F4).

Point 5:  Diagnostic accuracy of TyG. Given the availability of liver fibrosis staging, I suggest the authors to investigate the diagnostic accuracy of TyG index for the detection of significant fibrosis (F01 vs F234) and severe fibrosis (F012 - F34).

Response 5: Thank you for this suggestion, we made the required changes both in the manuscript body and Figure.

Point 6: Please provide footnotes for tables and figures, indicating the statistical test used, abbreviations, and all relevant information useful to readers.

Response 6: Thank you for pointing this out, we have added the requested footnotes.

Point 7: I suggest to provide subheadings in the Materials and Methods section.

Response 7: Thank you for the suggestion, we have added subheadings.

Round 2

Reviewer 1 Report

No further comments.

Author Response

We thank the reviewer for the comment. We are happy that the reviewer was pleased with the new version of the manuscript.

Reviewer 2 Report

The authors improved the manuscript according to almost all comments provided. Regarding point 5, can the authors provide the AUC values for the detection of significant fibrosis (F01 vs F234) and severe fibrosis (F012 vs F34)?

Author Response

We thank the reviewer for the comments. We are pleased that the reviewer finds that the new version of the manuscript was improved. We also thank you for the new suggestion. We performed the ROC curve for significant fibrosis, showing statistical significance. However, as the AUC value was not high enough in order to establish a cut-off for TyG index that could predict significant fibrosis.
